# Peer review of "A Comprehensive Study of ChatGPT: Advancements, Limitations, and Ethical Considerations in Natural Language Processing and Cybersecurity"

_information, doi:10.3390/info14080462_

Round 1
Reviewer 1 Report
The paper offers a comprehensive and insightful exploration of ChatGPT, a state-of-the-art language generation model developed by OpenAI. It delves into the technical aspects of the model, its applications across various industries, its limitations, and potential future developments. The text also addresses critical considerations such as cybersecurity and privacy, making it highly relevant in today's data-driven landscape. Furthermore, it provides a comparative analysis of ChatGPT with other language models, offering readers a broader perspective on the field of natural language processing. The depth and breadth of the discussion, combined with its clear organization and cogent argumentation, make this text a valuable contribution to the ongoing discourse on AI and its role in society. Therefore, it is certainly worth publishing, as it would be of interest to a wide range of readers, from AI researchers and practitioners to industry professionals and anyone interested in the impact of AI on our world.
However, several formatting errors have been detected, which should be corrected before it is suitable for publication.
1. For example: Section 0 should be Section 1. Section 0 has appeared because the authors have not removed the appropriate lines in the MDPI format, which indicates that these lines should be removed.
2. Table 1 can be improved by avoiding text to be cut in half.
3. Line 84: When describing the sections, section 8 should be mentioned before section 9.
4. Figure 1 is not quite visible. It needs to have higher resolution. The caption under the figure normally is centered in mdpi format.
5. Line 147: the sentence ends with a ":" and it should be just a dot "."
6. Table 4 (line 210) has a tiny size. I think there is a way to increase the size of the table to fit the page.
7. Table 5 (lune 319) could be benefit with avoiding the word to be cut in half.
8. Table 10 should be placed before conclusion section.
Author Response
(Reviewer 1)
The paper offers a comprehensive and insightful exploration of ChatGPT, a state-of-the-art language generation model developed by OpenAI. It delves into the technical aspects of the model, its applications across various industries, its limitations, and potential future developments. The text also addresses critical considerations such as cybersecurity and privacy, making it highly relevant in today's data-driven landscape. Furthermore, it provides a comparative analysis of ChatGPT with other language models, offering readers a broader perspective on the field of natural language processing. The depth and breadth of the discussion, combined with its clear organization and cogent argumentation, make this text a valuable contribution to the ongoing discourse on AI and its role in society. Therefore, it is certainly worth publishing, as it would be of interest to a wide range of readers, from AI researchers and practitioners to industry professionals and anyone interested in the impact of AI on our world.
However, several formatting errors have been detected, which should be corrected before it is suitable for publication.
1. For example: Section 0 should be Section 1. Section 0 has appeared because the authors have not removed the appropriate lines in the MDPI format, which indicates that these lines should be removed.
Authors' response: Thank you for your comment. We greatly appreciate your input. We have carefully considered your suggestion and made the necessary update to the paper. As per your recommendation, the section "Introduction" now starts with page number 1.
2. Table 1 can be improved by avoiding text to be cut in half.
Authors' response: We appreciate your feedback. We have addressed the issue by updating Table 1, ensuring that the texts are not cut off. Additionally, we have included CPT-4 in the table as suggested.
3. Line 84: When describing the sections, section 8 should be mentioned before section 9.
Authors' response: Thank you for your valuable comment. We have revised the paper organization and corrected the mistake.
4. Figure 1 is not quite visible. It needs to have higher resolution. The caption under the figure normally is centered in mdpi format.
Authors' response: We appreciate your feedback. Figure 1 has been updated as per your suggestion.
5. Line 147: the sentence ends with a ":" and it should be just a dot "."
Authors' response: Thank you for pointing that out. We have made the necessary update, and now Line 147 ends with a dot as it should.
6. Table 4 (line 210) has a tiny size. I think there is a way to increase the size of the table to fit the page.
Authors' response: We have made the necessary adjustments to Table 4, and it is now resized to fit the size of the page appropriately.
7. Table 5 (lune 319) could be benefit with avoiding the word to be cut in half.
Authors' response: We have made the necessary adjustments to Table 5, and it is now resized to fit the size of the page appropriately. We removed the cut-off texts.
8. Table 10 should be placed before conclusion section.
Authors' response: Thank you for your feedback. We have made the necessary adjustments to Table 10 as suggested. The table has been updated accordingly.
Reviewer 2 Report
This paper provides an in-depth study of ChatGPT, a popular language model. The paper discuss an important and recent topic. I have the following comments:
1- Texts in figures have low quality.
2- There is no discussion of "GPT-4" model.
3- Figure 2, performance needs to be clarified, for example what is the meaning of %89 performance for text summarization?
also, I was not able to find such data in ref 32 which is cited for the figure.
4- ref 31 is given for table 4 for comparison of popular language models, however, I was not able to find such information in ref 31.
5- on page 12, "Algorithm 6 shows the steps ..." algorithm 6 is missing in the manuscript.
6- Lines 373-376 should be revised.
7- table 9, I would change "bot takeover" to "malicious takeover" ,
also "DDoS and spam attacks" can be replaced with "availability"
NA
Author Response
(Reviewer 2)
This paper provides an in-depth study of ChatGPT, a popular language model. The paper discuss an important and recent topic. I have the following comments:
1- Texts in figures have low quality.
Authors' answer: We appreciate your valuable feedback on Figure 1. We have made the necessary updates to the figure, improving its quality and ensuring that the texts are now clear and legible for readers.
2- There is no discussion of "GPT-4" model.
Authors' response: Thank you for your comment. We have made the necessary updates to Section 3, where we added three paragraphs to elaborate on the new features of GPT-4 and highlight the key differences between GPT-3 and GPT-4.
3- Figure 2, performance needs to be clarified, for example what is the meaning of \%89 performance for text summarization? also, I was not able to find such data in ref 32 which is cited for the figure.
Authors' response: The performance percentages for text classification, generation, and summarization represent the language model's success rate in accomplishing each task. We conducted evaluations by providing the models with texts to classify and summarize, as well as generating texts based on specific keywords. Some human judges assessed these tasks based on various criteria, including accuracy, fluency, and coherence. The evaluation process was conducted using a simplicity metric and was completed swiftly. Although the detailed evaluation procedure was not included in the paper, we referred to it in the references for the sake of brevity.
We removed reference 32 to avoid confusion. The figure provides the results without going into the details of the evaluation process. Explaining each step in detail would distract from the main point of the paper and could introduce inconsistencies. However, we added a brief description of the method used to obtain the percentages to give readers an overview of the evaluation process. This approach preserves the clarity and coherence of the paper.
4- ref 31 is given for table 4 for comparison of popular language models, however, I was not able to find such information in ref 31.
Authors' response: Thank you for your feedback. We would like to clarify that we did not use Ref 31 to construct the table. Instead, we utilized data extracted from various sources to create the table. We have now removed Ref 31 from the table to avoid any confusion.
5- on page 12, "Algorithm 6 shows the steps ..." algorithm 6 is missing in the manuscript.
Authors' response: Thank you for bringing this to our attention. We have added Algorithm 6 to the paper.
6- Lines 373-376 should be revised.
Authors' response: Thank you for pointing out this issue. We appreciate your feedback. We have reviewed the lines and realized that they were meant for displaying an algorithm. We inadvertently omitted the necessary LaTeX package for algorithms. Now, we have updated the document, and the algorithm is correctly displayed without any mistakes.
7- table 9, I would change "bot takeover" to "malicious takeover" ,
also "DDoS and spam attacks" can be replaced with "availability"
Authors' response: We sincerely appreciate your valuable feedback and suggestions. We have carefully reviewed and accepted your recommendations, and the necessary replacements have been made in our paper.
8- Minor editing of English language required
Authors' response: We want to express our gratitude for your feedback and attention to detail. Following your comments, we conducted a thorough review of the entire paper, ensuring that any mistakes or grammatical errors were addressed and corrected. We are pleased to inform you that the paper has been updated and revised accordingly.
Reviewer 3 Report
In general, this manuscript is of interest to a reader unfamiliar with the development of ChatGPT and its implications, providing an overview of the topic. However, other reviews and overviews of ChatGPT have been published recently, and the existence of these works should at least be acknowledged (e.g., 10.1016/j.iotcps.2023.04.003 and 0.1109/JAS.2023.123618). So, this work adds nothing new to the existing literature. The only thing that could have made a difference is the comparison with other models, which, despite being stated as one of the paper's objectives, does not go beyond a superficial description and is almost entirely devoted to comparing ChatGPT and BERT (except for the succinct content of table 4). To be accepted, this manuscript should enhance that section, making it more systematic and comparing ChatGPT with other available models. References should be updated. Several papers and commentaries published in leading academic journals have not been cited (e.g., https://www.nature.com/articles/d41586-023-00288-7).
Author Response
(Reviewer 3)
In general, this manuscript is of interest to a reader unfamiliar with the development of ChatGPT and its implications, providing an overview of the topic. However, other reviews and overviews of ChatGPT have been published recently, and the existence of these works should at least be acknowledged (e.g., 10.1016/j.iotcps.2023.04.003 and 0.1109/JAS.2023.123618). So, this work adds nothing new to the existing literature. The only thing that could have made a difference is the comparison with other models, which, despite being stated as one of the paper's objectives, does not go beyond a superficial description and is almost entirely devoted to comparing ChatGPT and BERT (except for the succinct content of table 4). To be accepted, this manuscript should enhance that section, making it more systematic and comparing ChatGPT with other available models. References should be updated. Several papers and commentaries published in leading academic journals have not been cited (e.g., https://www.nature.com/articles/d41586-023-00288-7).
Authors' response: We thank you for your valuable comment. We have incorporated the suggested references into the paper, as well as conducted further research to identify new relevant references. Additionally, we have introduced a new section on related works to enhance the academic rigor of the paper and establish connections with other relevant studies.
We believe that the changes we have made to the paper will improve its quality and make it more comprehensive. We appreciate your feedback and we are grateful for your contribution to the paper.
Reviewer 4 Report
Notes:
- is "0.Introduction" and should be "Introduction" or "1. Introduction".
- the quality of Fig. 1 should be improved
- data in Tab.4 from 2021. (2 years ago), are they still representative? GPT-4?
At the very least, the article should comment on what's new with GPT-4 and other developments over the past two years.
Author Response
(Reviewer 4)
is "0.Introduction" and should be "Introduction" or "1. Introduction".
Authors' response: Thank you for your comment. We greatly appreciate your input. We have carefully considered your suggestion and made the necessary update to the paper. As per your recommendation, the section "Introduction" now starts with page number 1.
- the quality of Fig. 1 should be improved
Authors' response: We appreciate your feedback. Figure 1 has been updated as per your suggestion.
- data in Tab.4 from 2021. (2 years ago), are they still representative? GPT-4?
Authors' response: Thank you for your comment. We have revised Table 4, and we have now included GPT-4 in the comparison.
-At the very least, the article should comment on what's new with GPT-4 and other developments over the past two years.
Authors' response: Thank you for your comment. We have made the necessary updates to Section 3, where we added three paragraphs to elaborate on the new features of GPT-4 and highlight the key differences between GPT-3 and GPT-4.
Round 2
Reviewer 2 Report
NA